# Clinical outcomes of a community clinic-based lifestyle change program for prevention and management of metabolic syndrome: Results of the 'Vida Sana/Healthy Life' program

Patricia Markham Risica [1,2,3]* , Meghan L. McCarthy[4], Katherine L. Barry[4], Susan P. Oliverio[5‡], Kim M. Gans[1,2,6‡], Anne S. De Groot[4,7‡]

1 Center for Health Equity Research, Brown University School of Public Health, Providence, RI, United States of America, 2 Department of Behavioral and Social Sciences, Brown University School of Public Health, Providence, RI, United States of America, 3 Department of Epidemiology Brown School of Public Health, Providence, RI, United States of America, 4 Clínica Esperanza/Hope Clinic, Providence, RI, United States of America, 5 Department of Internal Medicine, Warren Alpert School of Medicine, Brown University, Providence, RI, United States of America, 6 Department of Human Development and Family Studies and Institute for Collaboration on Health, Interventions and Policy, University of Connecticut, Storrs, CT, United States of America, 7 Institute for Immunology and Informatics, University of Rhode Island, Providence, RI, United States of America

☽ These authors contributed equally to this work.
‡ These authors also contributed equally to this work.
* Patricia_Risica@brown.edu

## Abstract

### Introduction

As US Hispanic populations are at higher risk than non-Hispanics for cardiovascular disease and Type 2 diabetes targeted interventions are clearly needed. This paper presents the four years results of the Vida Sana Program (VSP), which was developed and is implemented by a small clinic serving mostly Spanish-speaking, limited literacy population.

### Methods

The eight-week course of interactive two-hour sessions taught by *Navegantes*, bilingual/cultural community health workers, was delivered to participants with hypertension, or high lipids, BMI, waist circumference, glucose or hemoglobin A1C (A1C). Measures, collected by *Navegantes* and clinic nurses, included blood chemistries, blood pressure, anthropometry, and an assessment of healthy food knowledge.

### Results

Most participants (67%) were female, Hispanic (95%), and all were 18 to 70 years of age. At baseline, close to half of participants were obese (48%), had high waist circumference (53%), or elevated A1C (52%), or fasting blood glucose (57%). About one third had high blood pressure (29%) or serum cholesterol (35%), and 22% scored low on the knowledge assessment. After the intervention, participants decreased in weight (-1.0 lb), BMI

**Data Availability Statement:** Data cannot be made publicly available as this data is from a small, vulnerable population, such that the information

provided in this dataset would potentially be identifiable due to very small numbers in some demographic groups. Data can be made available to any researcher or other professional that would like access to it. For further discussion of the data, please contact the Clinic Director of Clinica Esperanza/Hope Clinic, Morgan Leonard (contact via morganl@aplacetobehealthy.org or info@aplacetobehealthy.org).

**Funding:** This research was not funded by an external or intramural grant. No funders had a role in study design, data collection and analysis, decision to publish, or preparation of the manuscript.

**Competing interests:** The authors have declared that no competing interests exist.

(-0.2 kg/m2), WC (-0.4 inches), and cholesterol (-3.5 mg/dl, all p<0.001). Systolic blood pressure decreased (-1.7 mm Hg, p<0.001), and the knowledge score increased (6.8 percent, p<0.001).

## Discussion

VSP shows promising improvements in metabolic outcomes, similar to other programs with longer duration or higher intensity interventions. VSP demonstrates an important model for successful community-connected interventions.

## Introduction

Hispanic populations in the United States are at higher risk of behavioral and metabolic risk factors and experience lower access to health screening and healthcare services than other racial or ethnic groups in the United States [1]. Hispanic populations are also at higher risk for developing cardiovascular disease [2] and Type 2 diabetes due to the confluence of certain risk factors in the population [3]. For example, Hispanics have the highest risk among US racial/ethnic groups for developing Metabolic Syndrome (MetS), a condition combining hyperglycemia/insulin resistance, obesity and dyslipidemia [4–6]. In particular, according to 2007–2012 data, Mexican American women and men have higher fasting blood glucose compared with Non-Hispanic Whites (NHW), and Mexican American women also have higher risk of high waist circumference and lower HDL than NHW women [6]. Higher risk of MetS and Type 2 diabetes for Hispanic women in the Northeastern US compared to NHW women has been documented [1]. These disparities clearly highlight a need for targeted intervention, though what kind of intervention and which behaviors to target is less clear.

Several important behavioral risk factors associated with MetS that might be natural targets for intervention have been measured with US Hispanics as previously described [7]. Smoking and Alcohol consumption are lower in this population than for NHWs [8]. Initial dietary habits of recent Hispanic immigrants are healthier than NHWs, but change to be less-healthy, with acculturation [8, 9], which is consistent with findings other research highlighting higher consumption of sugar sweetened beverages, and fewer fruits and vegetables among Hispanic, compared to NHW Americans [10]. Additionally, sedentary behaviors and physical inactivity are higher among Hispanic Americans compared to NHW [8]. These dietary and physical activity behaviors may be important factors to address in lifestyle intervention programs.

Community-based interventions have the potential to inform and engage community members at risk for MetS-associated diseases about how to manage their diseases and modify their diet and exercise regimens, which may result in lower risks of morbidity and cost of care. Clínica Esperanza/Hope Clinic (CEHC), located on the West Side of Providence, Rhode Island, serves a low-income, often uninsured, largely Spanish-speaking, and literacy-limited population. CEHC personnel developed a lifestyle intervention called the *Vida Sana Program* (VSP) to engage patients and community members in the prevention and treatment of MetS and related diseases by encouraging healthier lifestyles [11]. A previous manuscript reported data for the pilot year of the VSP intervention at CEHC [12]. Data for years 2–5 of VSP with enrollment dates between January 2014 and June 2017 are presented here, documenting the clinical outcomes (n = 641) and providing support for the implementation of culturally sensitive, linguistically appropriate programs for low health literacy populations.

## Methods

The methods utilized in VSP have been documented in detail previously [11]. The protocol for this work was reviewed by, "Ethical and Independent Review Services," in April 2016, then annually since that time. The IRB determined that this protocol (18021–01) meets the criterion for exemption. The data collected for this manuscript was not from the medical charts, but was collected independently for evaluation of Vida Sana. Researchers only had access to deidentified data for the purposes of this analysis. In brief, VSP was designed to serve the mostly low health-literate and reading-limited clinic population, to be sensitive to the cultural, social and linguistic factors that affect CEHC patients and to enable patients and community members to take ownership of their own health. The interactive eight-week course is taught by trained bilingual/bi-cultural community health workers called *Navegantes* who have an ongoing relationship with many CEHC patients as they also provide a source of support in navigating the healthcare system in other roles within CEHC.

*Navegantes* are available during all clinical encounters CEHC, serving as liaisons for most patients between patients and the healthcare system by offering support relating to insurance options, specialty referrals, and social determinants of health. They are specifically trained to provide nutritional and chronic disease information and advice, which may include offering enrollment into the Vida Sana Program and can act as advocates for the individual patients navigating the healthcare system and the clinic.

### Recruitment of participants and eligibility criteria

CEHC medical providers recruit patients into VSP if they have or are at risk for hypertension, have elevated blood lipids, BMI, waist circumference, fasting blood glucose or Hemoglobin A1C (A1C). Specific and detailed guidelines for inclusion or exclusion are not applied; providers are able to refer any patients who they determine to be at risk for metabolic syndrome or who would benefit from the program. Recruitment also extends to community members who seek out the VSP program after word-of-mouth information from other participants. Most groups take place at the clinic itself, but non-CEHC participants are also recruited into VSP classes when a class is planned in another location accessible to the clinic population, such as community sites and religious centers. As VSP is a clinic-based program, not a research study, we did not calculate an apriori sample size based on anticipated power, but included in this analysis all participants enrolled during the stated time frame.

VSP incentivizes attendance at weekly sessions by offering grocery store gift cards and healthy snacks. One $10 gift card is given out upon enrollment, another at the week 8 session, and a third at their 12-week appointment to check clinical measures. We encourage participants to use these gift cards to buy ingredients to make the healthy recipes that are demonstrated during class. Volunteers at the clinic offer childcare during the classes in order to help parents of young children attend.

### Intervention

VSP consists of one- to two-hour group sessions over 8 weeks delivered by *Navegantes* in small groups (10–18 participants) in English and Spanish, following a standard VSP curriculum containing specific learning objectives for each session. The materials called Thumbs Up™ includes workbooks and presentation materials and facilitates learning through simple, easy-to-understand terminology and vivid imagery. More details about the intervention, including recruitment and training of *Navegantes* are published elsewhere [11]. The groups involve approximately 70% discussion and 30% sharing of information. Content of the sessions includes information about the health conditions and related clinical indicators associated

with Type 2 diabetes, hypertension and hypercholesterolemia, along with practical knowledge and skills associated with improving diet, physical activity, medication compliance, and decreasing sedentary time. During the week between sessions, the *Navegante* reaches out to each participant by telephone to remind them of the location, confirm their attendance and reinforce their session content and their personal goals. The *Navegantes* create a mutually supportive environment similar to a "social club" for participants through the use of group games and story-telling. Through interactive learning and celebration of participants' achievements with certificates, the *Navegantes* create a group environment in which participants can share stories, empathize about challenges and celebrate successes.

## Evaluation measures

*Navegantes* and CEHC nurses conducted all measures during the first session attended by the participant. The measures include anthropometries, blood chemistries and an assessment of diet, health and wellness knowledge.

- Height, Weight and BMI: Participants are weighed without shoes on a scale dedicated to the VSP. Height is measured on the stadiometer attached to the dedicated scale, also without shoes. BMI is calculated using these measurements as weight (kg) divided by height (meters) squared.

- Fasting Blood Glucose: Blood glucose testing is conducted using the Assure Platinum 7-Second Blood Glucose Meter [13]. Participants were asked to fast for at least four hours prior to the blood draw.

- Hemoglobin A1C (A1C): Testing is done using the Alere Afinion, which offers accurate results from 1.5 μL of finger stick or venous whole blood [14].

- Cholesterol: Fasting cholesterol is tested using the mobile Alere Cholestech LDX [15].

- Blood Pressure: Systolic and diastolic blood pressure measurements are taken by trained staff members using a calibrated sphygmomanometer that is dedicated to VSP. Blood pressure is measured twice, with at least five minutes between measurements. If there is a significant difference between the first two readings, blood pressure is measured again at the end of the session, and this value is used.

- Waist Circumference: A dedicated paper measuring tape is used to measure patients' waist circumference just above the hip bones, at the widest point.

Follow-up measures were taken at the end of the intervention (8 weeks). (A second round of follow-up measures were taken at 12 weeks after the intervention start for any participants who were able to come back, but those data are not included.) Participants also complete a pre- and post-intervention knowledge assessment that tests their recognition of healthy food labels, food choices and clinical numbers. Scores for this tool are presented as a percentage of the 20 items presented. The *Navegantes* administer the test verbally, using PowerPoint slides with graphics and photos to illustrate the questions, while participants record their responses on an answer sheet. This assessment consists of several tasks: 1) choosing which is the healthier of two food options, 2) classifying blood pressure readings as "healthy" or "unhealthy", and 3) comparing the salt or sugar content of two different foods or food labels.

## Data analysis

To assess baseline risk for each blood chemistry, anthropometry or blood pressure value, high risk was defined as: BMI>30 kg/m$^2$ [16], waist circumference >35 cm (women) or WC >40

cm (men) [16], systolic blood pressure >130 mmHg, diastolic blood pressure > 80 mmHg [17], A1C >5.6 g/dl [18], glucose >100 mg/dl [19], cholesterol >200 mg/dl [20]. Demographics and baseline clinical risk variables were compared between those with and without follow-up data using chi square (Table 1). Differences in baseline clinical variables were examined by demographics of age group, sex and primary language spoken with ANOVA models (Table 2). Change in each measure was assessed with paired t-tests for the sample with complete pre and post data as measured (Table 3). Also change in each measure was assessed for just participants identified as having high risk values at baseline, also using paired t-tests (Table 3). Missing follow-up data were imputed by carrying forward the baseline value for each participant (also Table 3). Changes between baseline and Week 8 data were assessed by demographic characteristics for both crude (complete cases only) and for imputed data using ANOVA models.

## Results

Of the 641 enrolled VSP participants at baseline (Table 1), 399 (62%) completed the program and were assessed at the end of the 8 weeks. At baseline, participants were well distributed among age groups from 18 to over 65. Most participants (67%) were female, Hispanic (95%), and Spanish speaking (92%). Similar proportions reported being publically insured (27%) or uninsured (32%), with a large group (32%) not responding, and fewer (9%) reporting being privately insured. Participants identified their country of origin as Guatemala (34.9%), Dominican Republic (28.4%), Puerto Rico (8%), Columbia (4.4%) or the US (3.7%) with all others (Mexico, Bolivia, El Salvador, Honduras, Venezuela, Peru, Argentina, Chile, Cuba, Brazil, Haiti, Philippines, Burundi, Cape Verde, Gibralter and Senegal) reporting in lower percentages, and 3.9% not responding (Data not shown).

Many participants had baseline clinical characteristics that are consistent with chronic disease risk. Over half of participants (48%) were obese, with another third (34%) overweight, with only 18% having normal weight status. Just over half (53%) had waist circumference consistent with being high risk. More than half (52%) of participants had A1C levels that are consistent with some degree of risk (>5.6%), which is consistent with the finding that over half (57%) had fasting blood sugar greater than 100mg/dl. Just over one quarter participants had elevated baseline blood pressure (29% for systolic (>130 mmHg) and 28% for diastolic (>80 mmHg). Over one third (35%) of participants had elevated cholesterol at baseline. Just under one quarter of participants (22%) scored lower than 60% on the knowledge assessment.

Demographics and clinical characteristics were compared between those with and without follow-up data to assess differential dropout, which might indicate a bias of the assessed data toward certain characteristics. A higher proportion of women completed the program and final evaluation than men (p<0.01). Also, people with normal A1C (p<0.05) and glucose (p<0.05) were more likely to completed the program compared with those with elevated A1C or glucose. No other differences in demographic or clinical characteristics were found between those who did and did not complete the program.

Many of the baseline clinical measures differed by demographic characteristics (Table 2). All clinical chemistry measures and anthropometries differed by age group, but not always in a linear pattern. Other differences were found by sex, race or ethnicity.

Changes in clinical values for all participants were mixed. Table 3 shows outcomes from the crude data set with only those participants who had baseline and follow-up data (Fig 1), as well as the full dataset including imputation for missing values. Many changes observed for those with complete case were found to be diminished with imputation for missing values. Results described here focus on the outcomes after imputations. At the completion of the 8-week intervention, participants averaged reductions in weight (1.0 lb), BMI (0.2 kg/m2), and WC

**Table 1. Demographic and metabolic characteristics of participants by status of completing the VSP program.**

| | All | | Completed | | Not Completed | | P Value |
|---|---|---|---|---|---|---|---|
| | Total: 641 | | Total: 399 (62%) | | Total: 242 (38%) | | |
| **Age** | n | % | n | % | n | % | p = 0.26 |
| 18–34 years | 124 | 19.3% | 68 | 17.0% | 56 | 23.1% | |
| 35–44 years | 130 | 20.3% | 81 | 20.3% | 49 | 20.2% | |
| 45–54 years | 137 | 21.4% | 91 | 22.8% | 46 | 19.0% | |
| 55–64 years | 114 | 17.8% | 75 | 18.8% | 39 | 16.1% | |
| 65+ years | 123 | 19.2% | 81 | 20.3% | 41 | 16.9% | |
| Unreported | 13 | 2.0% | 3 | 0.8% | 11 | 4.5% | |
| **Sex** | n | % | n | % | n | % | p < .01 |
| Male | 198 | 30.9% | 105 | 26.3% | 93 | 38.4% | |
| Female | 431 | 67.2% | 292 | 72.9% | 139 | 57.4% | |
| Unreported | 12 | 1.9% | 3 | 0.8% | 10 | 4.1% | |
| **Ethnicity** | n | % | n | % | n | % | p = 0.42 |
| Hispanic | 607 | 94.7% | 384 | 96.2% | 223 | 92.1% | |
| Not Hispanic | 9 | 1.4% | 4 | 1.0% | 5 | 2.1% | |
| Unreported | 25 | 3.9% | 11 | 2.8% | 14 | 5.8% | |
| **Primary Language** | n | % | n | % | n | % | p = .63 |
| Spanish | 589 | 91.9% | 372 | 93.2% | 217 | 89.7% | |
| English | 25 | 3.9% | 16 | 4.0% | 9 | 3.7% | |
| Other[1] | 5 | 0.8% | 5 | 1.3% | 0 | 0.0% | |
| Unreported | 22 | 3.4% | 6 | 1.5% | 16 | 6.6% | |
| **Insurance Status** | n | % | n | % | n | % | p = 0.16 |
| Insured (Public) | 175 | 27.3% | 93 | 23.3% | 43 | 17.8% | |
| Insured (Private) | 58 | 9.0% | 41 | 10.3% | 17 | 7.0% | |
| Uninsured | 203 | 31.7% | 122 | 30.6% | 81 | 33.5% | |
| Unreported | 205 | 32.0% | 143 | 35.8% | 101 | 41.7% | |
| **Weight Status** | n = 621 | | n = 399 | | n = 222 | | |
| | n | % | n | % | n | % | 0.844 |
| Normal or underweight ($<25$ kg/m$^2$) | 114 | 18.4 | 74 | 18.5 | 40 | 18.0 | |
| Overweight ($>25$ and $<30$ kg/m$^2$) | 209 | 33.7 | 131 | 32.8 | 78 | 35.1 | |
| Obese ($>30$ kg/m$^2$) | 298 | 48.0 | 194 | 48.6 | 104 | 46.8 | |
| **Waist Circumference[2]** | n = 600 | | n = 390 | | n = 210 | | |
| | n | % | n | % | n | % | |
| Normal <br> < = 44 inches (men) <br> < = 34 inches (women) | 284 | 47.3 | 177 | 45.4 | 107 | 51.0 | 0.2236 |
| High <br> >44 inches (men) <br> >34 inches (women) | 316 | 52.7 | 213 | 54.6 | 103 | 49.0 | |
| **Hemoglobin A1C** | n = 240 | | n = 153 | | n = 87 | | |
| | n | % | n | % | n | % | |
| Normal ($< 5.6\%$) | 113 | 47.1 | 78 | 51.0 | 35 | 40.2 | 0.048 |
| $>5.6 < 7.0\%$ | 87 | 36.3 | 57 | 37.3 | 30 | 34.5 | |
| $>7.0 < 9.0\%$ | 27 | 11.3 | 13 | 8.5 | 14 | 16.1 | |
| $>9.0\%$ | 13 | 5.4 | 5 | 3.3 | 8 | 9.2 | |
| **Glucose** | n = 616 | | n = 398 | | n = 218 | | |
| | n | % | n | % | n | % | |

*(Continued)*

**Table 1.** (Continued)

| | All | | Completed | | Not Completed | | P Value |
|---|---|---|---|---|---|---|---|
| | **Total: 641** | | **Total: 399 (62%)** | | **Total: 242 (38%)** | | |
| Normal (< = 100 mg/dl) | 266 | 43.2 | 187 | 47.0 | 79 | 36.2 | 0.0128 |
| Elevated (>100 mg/dl) | 350 | 56.8 | 211 | 53.0 | 139 | 63.8 | |
| **Systolic Blood Pressure** | n = 619 | | n = 398 | | n = 221 | | |
| | n | % | n | % | n | % | |
| Normal (< = 130 mmHg) | 440 | 71.1 | 289 | 72.6 | 151 | 68.3 | 0.721 |
| High (>130 mmHg) | 179 | 28.9 | 109 | 27.4 | 70 | 31.7 | |
| **Diastolic Blood Pressure** | n = 618 | | n = 397 | | n = 221 | | |
| | n | % | n | % | n | % | |
| Normal (< = 80 mmHg) | 448 | 72.5 | 285 | 71.8 | 163 | 73.8 | 0.341 |
| High (>80 mmHg) | 170 | 27.5 | 112 | 28.2 | 58 | 26.2 | |
| **Cholesterol** | n = 606 | | n = 389 | | n = 217 | | |
| | n | % | n | % | n | % | |
| Normal (< = 200 mg/dl) | 391 | 64.5 | 258 | 66.3 | 133 | 61.3 | 0.245 |
| High (>200 mg/dl) | 215 | 35.5 | 131 | 33.7 | 84 | 38.7 | |
| **Knowledge**[3] | n = 470 | | n = 342 | | n = 128 | | |
| | n | % | n | % | n | % | |
| > 60% | 367 | 78.1 | 273 | 79.8 | 94 | 73.4 | 0.1723 |
| < = 60% | 103 | 21.9 | 69 | 20.2 | 34 | 26.6 | |

Notes

[1]Waist Circumference is assessed as high/low separately for men and women.

[2]Other language includes Portuguese, Haitian Creole.

[3]Knowledge is presented as percentage correct of 20 items.

(0.4 inches) (p<0.001 for all). More than 5% of baseline weight was lost for 5.5% of the sample (data not shown). No significant changes were found for A1C or blood glucose levels. Systolic blood pressure decreased by 1.7 mm Hg (p<0.001), but no significant change in diastolic blood pressure was found. Cholesterol averaged a decrease of 3.5 mg/dl (p<0.001). The knowledge score increased overall by 6.8 percent. (p<0.001).

When assessing changes in only the participants with high-risk levels of each clinical value at baseline, improvements were found in all measures (Table 3). Decreases of 0.8 lb (weight), 0.2 kg/m2 (BMI), and 0.5 inches (WC, all p<0.001) were very similar to the overall group changes. A1C and blood glucose both decreased (-0.3, p<0.01 and -8.3 mg/dl, p<0.001 respectively). Both systolic and diastolic blood pressure decreased (-6.0 mm Hg p<0.001 and -5.1 mm Hg respectively, p<0.001). Total cholesterol levels decreased by 12.1 mg/dl (p<0.001). The knowledge score among those with initial low values increased by an average of 18.8 points (p<0.001).

Changes in outcomes were also examined by age group, sex and language. No differences by demographic characteristics were found for crude changes and those with imputation of missing values.

## Discussion

VSP, a low-intensity, eight-week program delivered by CHWs (*Navegantes)* resulted in reductions in clinical risk factors, especially among participants with high risk levels at baseline. VSP recruited mostly Hispanic participants, which is to be expected from a clinic serving a mostly

**Table 2. Baseline clinical measures (mean (SD)) by age group, sex and language.**

| | Baseline values by demographic characteristics. | | | | | | | | | | | |
|---|---|---|---|---|---|---|---|---|---|---|---|---|
| | Age Group | | | | | | Sex | | | Language | | |
| | 18–34 (n = 124) | 35–44 (n = 130) | 45–54 (n = 137) | 55–64 (n = 114) | 65 and up (n = 123) | p value | Male (n = 191) | Female (n = 425) | p value | Spanish (n = 578) | English (n = 25) | p value |
| **Weight (lbs)** | 161.7 (38.0) | 171.8 (50.0) | 181.3 (40.4) | 168.3 (31.7) | 162.3 (28.2) | <0.001 | 178.4 (39.2) | 165.2 (38.7) | <0.001 | 167.8 (35.0) | 203.2 (88.0) | <0.001 |
| **BMI (kg/m$^2$)** | 29.3 (7.3) | 31.1(9.1) | 33.8 (7.9) | 31.5 (6.9) | 30.9 (6.0) | <0.001 | 30.6 (7.2) | 31.7 (7.9) | 0.093 | 31.2 (7.1) | 35.2 (16.3) | <0.05 |
| **WC (inches)** | 35.6 (5.3) | 37.5 (5.5) | 39.1 (5.4) | 38.5 (4.4) | 39.3 (4.5) | <0.001 | 38.4 (4.5) | 37.8 (5.5) | 0.206 | 37.9 (5.1) | 39.8 (7.2) | 0.076 |
| **AIC (mg/dl)** | 5.5 (1.4) | 5.8 (1.2) | 6.5 (1.8) | 6.1 (1.0) | 6.7 (1.7) | <0.01 | 6.4 (1.9) | 5.9 (1.2) | <0.05 | 6.1 (1.6) | 5.5 (0.62) | 0.134 |
| **Glucose (mg/dl)** | 105.1 (38.8) | 119.5 (59.6) | 127.0 (61.7) | 119.1 (42.8) | 133.3 (62.4) | <0.01 | 130.0 (66.4) | 116.9 (48.6) | <0.01 | 121.8 (56.3) | 101.6 (22.5) | 0.081 |
| **Systolic Blood Pressure (mmHg)** | 119.7 (12.1) | 122.2 (12.1) | 125.2 (13.2) | 129.1 (15.8) | 132.0 (16.1) | <0.001 | 127.6 (13.6) | 124.5 (14.8) | <0.05 | 125.6 (14.5) | 121.9 (14.3) | 0.211 |
| **Diastolic Blood Pressure (mmHg)** | 77.4 (9.8) | 80.3 (8.1) | 81.1 (13.9) | 79.8 (11.3) | 77.0 (8.9) | <0.01 | 80.0 (10.9) | 78.7 (10.6) | 0.190 | 78.9 (10.6) | 80.8 (12.5) | 0.390 |
| **Cholesterol (mg/dl)** | 182.1 (38.1) | 191.0 (37.8) | 196.4 (39.3) | 194.4 (38.6) | 183.3 (38.7) | <0.01 | 191.9 (41.0) | 188.5 (37.8) | 0.314 | 190.6 (39.0) | 168.0 (33.3) | <0.01 |
| **Knowledge Assessment[1]** | 79.2 (15.6) | 78.0 (17.0) | 75.5 (20.6) | 75.5 (18.4) | 69.0 (20.0) | <0.01 | 72.2 (16.8) | 77.0 (19.2) | <0.05 | 75.2 (18.7) | 86.2 (13.2) | <0.01 |

Notes

[1]Knowledge is presented as percentage correct of 20 items.

Hispanic clientele. Participants representing a wide age distribution were enrolled, but as observed for many health promotion interventions [21–23], women were more likely to participate and complete the 8-week program than men.

Many participants enrolled in VSP were at high risk with regard to weight status (34% overweight and 48% obese) and body fat distribution (53% had high WC). Additionally, higher-risk glucose intolerance was found within this population based on glucose (57% elevated) and A1C (53% elevated). However, fewer participants had high blood pressure (29% elevated systolic, 28% elevated diastolic) or blood cholesterol (36% elevated). This pattern of risks among a clinic sample serving mostly Spanish-speaking Hispanic participants is consistent with those found in other research [6, 24]. Often "Hispanic" populations are depicted based on data from Mexican American groups [6, 24], though Spanish-speaking populations from Caribbean, Central and South American origin as in VSP may have different profiles [25]. It is also likely that the risk pattern of program participants represents the clinical criteria by which CEHC clinicians referred patients to the VSP program.

Outcomes from VSP were comparable to similar studies aimed at addressing cardiometabolic risk factors [22, 23, 26–28]. However, changes in weight and BMI were the least noteworthy of the outcomes from VSP. The mean reduction (after imputation) in weight of 1.0 lb and BMI by 0.2 kg/m$^2$ is considerably more modest that that found by others. Diabetes Prevention Program (DPP) participants, which received a much more intensive 24 week intervention averaged a decrease of 12.3 lbs [26], and a 13-week version of DPP modified for Latinos achieved a weight decrease of 2.5 lbs [27]. Vivamos Activos also saw a decrease of 4.6 lbs and a decrease in BMI by 0.7 kg/m2 after the 6-month intervention delivered by CHWs [28]. No significant decrease in BMI was found in the "Latinos en Control" program, which included 12 weeks of an intensive phase (weekly sessions), and 8 months of monthly sessions [23].

Changes in BMI for VSP might also be modest as the follow-up time period was fairly short, and weight loss was not strongly encouraged as a goal for the program, as opposed to

**Table 3. Changes in clinical measures and chemistries (mean (SD)) from baseline to week 8 with baseline values imputed for missing.**

| | Complete Case | | | | | | | | |
| --- | --- | --- | --- | --- | --- | --- | --- | --- | --- |
| | All Participants | | | | Baseline High Risk Only | | | | |
| | n | Baseline | 8 weeks | Change | P value | n | Baseline | 8 weeks | Change | P value |
| Weight (lb) | 399 | 169.5 (41.8) | 168.0 (41.9) | -1.51 (4.4) | < .001 | 194 | 193.2 (44.7) | 191.7 (44.9) | -1.5 (5.0) | <0.001 |
| BMI (kg/m$^2$) | 392 | 31.6 (8.2) | 31.4 (8.2) | -0.3 (0.8) | < .001 | 187 | 37.8 (7.8) | 37.5 (7.9) | -0.3 (1.0) | <0.001 |
| Waist Circumference (inches) | 388 | 38.1 (5.2) | 37.4 (5.4) | -0.7 (2.0) | < .01 | 210 | 40.6 (4.5) | 39.9 (5.1) | -0.8 (2.4) | <0.001 |
| Hemoglobin A1C (%) | 133 | 5.91 (1.2) | 5.76 (0.86) | -0.15 (0.7) | < .05 | 56 | 6.8 (1.5) | 6.4 (1.0) | -0.39 (1.0) | <0.01 |
| Glucose (mg/dl) | 392 | 115.9 (46.1) | 112.8 (39.4) | -3.1 (40.0) | 0.13 | 205 | 140.3 (52.3) | 126.2 (47.4) | -14.1 (48.9) | <0.001 |
| Systolic Blood Pressure (mm Hg) | 397 | 125.6 (14.6) | 122.9 (13.6) | -2.7 (12.6) | < .01 | 108 | 144.6 (10.1) | 134.9 (12.6) | -9.9 (14.8) | <0.001 |
| Diastolic Blood Pressure (mm Hg) | 396 | 78.9 (10.1) | 78.1 (9.8) | -0.7 (11.3) | 0.20 | 112 | 90.1 (7.8) | 82.5 (8.4) | -7.7 (10.6) | <0.001 |
| Cholesterol (mg/dl) | 371 | 188.8 (38.3) | 183.0 (36.1) | -5.8 (27.8) | < .01 | 125 | 231.2 (23.3) | 210.4 (35.2) | -20.7 (30.6) | <0.001 |
| Knowledge | 303 | 75.6 (18.6) | 85.9 (13.9) | 10.6 (17.1) | < .001 | 61 | 45.3 (11.7) | 77.1 (12.4) | 31.8 (18.6) | <0.001 |
| | Imputing Baseline for Missing Follow-Up Values | | | | | | | | |
| | All Participants | | | | Baseline High Risk Only | | | | |
| | n | Baseline | 8 weeks | Change | P value | n | Baseline | 8 weeks | Change | P value |
| Weight (lb) | 622 | 169.0 (39.3) | 168.0 (39.4) | -1.0 (3.6) | < .001 | 285 | 191.3 (40.4) | 190.5 (40.6) | - 0.8 (3.9) | <0.001 |
| BMI (kg/m$^2$) | 608 | 31.3 (7.7) | 31.1 (7.6) | -0.2 (0.7) | < .001 | 283 | 37.3 (7.0) | 37.1 (7.0) | -0.2 (0.8) | <0.001 |
| Waist Circumference (inches) | 606 | 38.0 (5.2) | 37.5 (5.4) | -0.4 (1.6) | < .001 | 316 | 40.6 (4.6) | 40.1 (5.0) | -0.5 (2.0) | <0.001 |
| Hemoglobin A1C (g/dl) | 200 | 6.1 (1.5) | 6.0 (1.3) | -0.2 (0.6) | 0.179 | 87 | 7.1 (1.8) | 6.8 (1.7) | -0.3 (0.9) | <0.01 |
| Glucose (mg/dl) | 616 | 120.9 (54.9) | 118.9 (51.7) | -2.0 (32.0) | 0.131 | 350 | 145.0 (62.5) | 136.7 (61.2) | -8.3 (38.0) | <0.001 |
| Systolic Blood Pressure (mm Hg) | 618 | 125.5 (14.5) | 123.8 (13.9) | -1.7 (10.2) | < .001 | 179 | 143.9 (9.7) | 137.9 (11.8) | -6.0 (12.5) | <0.001 |
| Diastolic Blood Pressure (mm Hg) | 619 | 79.1 (10.7) | 78.6 (10.5) | -0.5 (9.0) | 0.196 | 170 | 90.7 (10.7) | 85.7 (11.8) | -5.1 (9.3) | <0.001 |
| Cholesterol (mg/dl) | 606 | 189.7 (38.8) | 186.1 (37.7) | -3.5 (21.9) | < .001 | 215 | 231.0 (25.7) | 218.9 (34.2) | -12.1 (25.4) | <0.001 |
| Knowledge | 470 | 75.6 (18.6) | 82.4 (16.7) | 6.8 (14.6) | < .001 | 103 | 45.8 (11.6) | 64.7 (22.3) | 18.8 | < .001 |

Notes

[1]Knowledge is presented as percentage correct of 20 items.

increasing health literacy, and overall healthier food choices and more physical activity. This is in contrast to other compared studies including DPP, which aimed for seven percent weight loss [26], and Vivamos Activos, which focused on tailored weight loss [28]. Like VSP, the Latinx modified DPP focused on healthy food choices, and, "Latinos en Control," which focused on diabetes knowledge and healthy food choices [23].

Waist circumference, a key indicator of MetS, is not often included in intervention studies focused on cardiometabolic factors. WC decreased by 0.4 inches among the entire VSP group, and slightly more (0.5 inches) among those starting CSP with an elevated WC. In the HELP Prevent Diabetes Trial, adapted from DPP and focusing on weight loss, participants achieved larger decreases with a loss of 2.6 inches at 12 months, which were sustained to 1.6 inches at 24 months [22]. No change in WC was reported [23, 26, 27] or WC was not measured [28] for other cardiometabolic studies reviewed here.

Changes in indicators of diabetes risk or metabolic syndrome among VSP participants compare quite favorably with other similar, but longer and more intensive interventions. A1C declined by 0.2g/dl for all VSP participants, 0.3 g/dl for the high-risk group, which was a slightly smaller improvement than for the more intensive DPP adapted for Latinos (a between group decrease in blood glucose of 0.5 g/dl at 4 months) [27]. Among those beginning VSP with elevated blood glucose levels, a reduction of 8.3 mg/dL in blood glucose was observed, which was slightly greater than the changes seen in the HELP Prevent Diabetes Trial, where participants averaged a significant decrease of 2.9 mg/dl at 12 months and 4.4 mg/dl at 24 months [22].

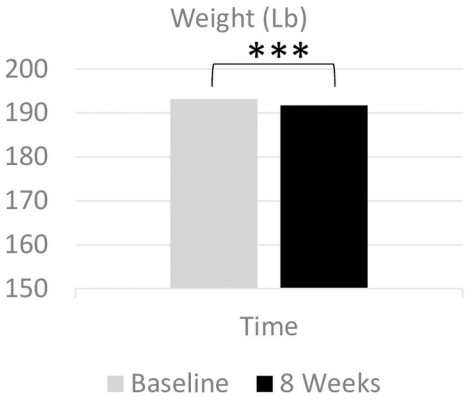

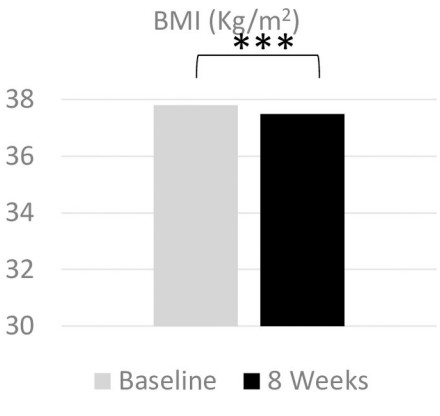

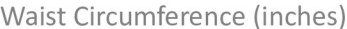

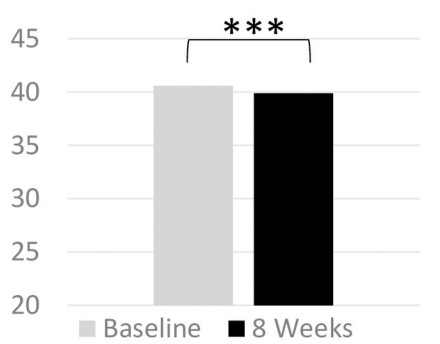

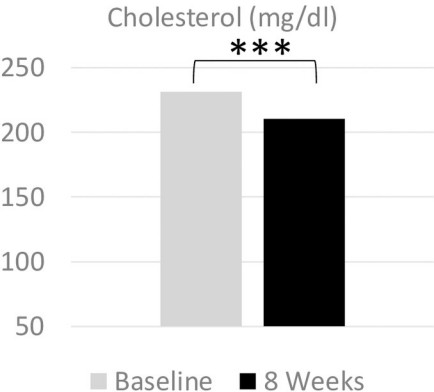

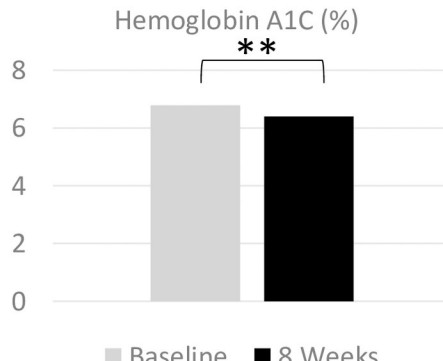

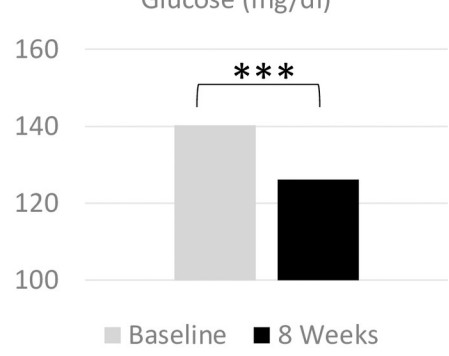

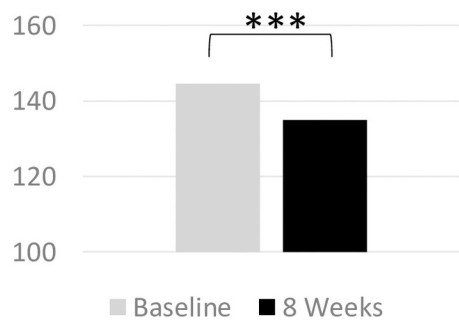

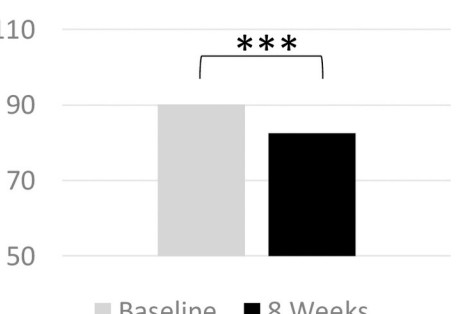

**Fig 1. Changes in clinical and chemical indicators for participants with high risk baseline values from baseline to 8 weeks post Vida Sana initiation.**

Few cardiometabolic-focused interventions have assessed lipid levels. Among VSP participants, total cholesterol levels improved (-3.5 for all and -12.1 for those at high risk), which was similar to or better than other programs. The Vivamos Activos program found small decreases (<5 mg/dl) among all study groups at 6 months, but no differences between groups [28]. Others did not measure total cholesterol [22, 27], but saw improvements in triglyceride and HDL levels [22], while others saw no significant intervention effect on lipids [23, 29].

Few studies addressing hyperlipidemia risk have also targeted metabolic risk or blood pressure. Blood pressure improved minimally among the whole sample, but significantly with a drop of 6.0 mmHg systolic and 5.1 mmHg for diastolic among the high-risk VSP participants. Other studies have found minimal [22] or no change [29], or did not measure blood pressure change.

In addition to clinical measures and chemistries, VSP also assesses knowledge of concepts related to nutrition and clinical indicators for chronic disease. The scores of this assessment improved substantially throughout the program, with the general group of VSP participants scoring 10 points higher at the Week 8 time point, and a more than 30-point increase observed among those with lower scores to start. This measure is particularly useful as it is conducted using pictures and symbols that require minimal literacy skills; however, mean scores at baseline were already relatively high indicating little room for improvement in knowledge. Few other studies have examined changes in health knowledge, especially in this way. The Latina Preventive Heart Disease Program conducted an assessment of CVD-related knowledge, including knowledge of heart attack symptoms, when to call 911, and CVD risk factors; they found significant increases in knowledge between the pre- and post-intervention survey [30]. Using the "Audit of Diabetes Knowledge," [31] the Latinos En Control intervention study showed significant increases in participants' knowledge at 4 months and 1-year post-intervention as compared with patients receiving usual care [23].

The lower weight reduction attained by VSP may be due to the shorter time frame of the intervention, the lower intensity of the intervention, or different content (less focused on weight loss) compared with the other studies reviewed. Though clinically significant weight reduction was not appreciably achieved in VSP, the changes in glucose, cholesterol and blood pressure were comparable or better than other more intensive studies, demonstrating that some behavioral change has occurred. The VSP study did not measure reported changes in physical activity or diet, which may or may not have occurred. Also, it is possible that behavioral change may have occurred in the form of medication compliance, which was also not measured. Increasing the duration or intensity of the short VSP program might result in better results, including more weight change, but those changes would be at the risk of lower retention due to increased participant burden.

Of the participants present for the baseline evaluation, 62% were present for the final class evaluation. Of the initially enrolled sample, 70% attended four or more classes (half of the course), but only 30% attended all eight classes. The retention rate, though not extraordinary, is comparable to others that ranged from 27.6–96% in a review of 9 RCTs that targeted Latino adults (mostly only women) [32]. The VSP study provided a $10 incentive at baseline and at 8 and 12 week follow-up evaluation time points; the DPP adapted for Hispanic adults, which offered an incentive of $25 at baseline and 6 months evaluations, and $50 at one year, reported a 1-year retention of 94% [27]. The HELP Prevent Diabetes Trial saw 92% retention at 6-months and 84% retention at 24-months [22]. Vivamos Activos saw a median of 12 sessions

attended out of 16 (75%) attended [28]. In the Latinos en Control program, 68% of participants attended more than 6 of the 12 total sessions [23]. These are comparable to the attendance for VSP. VSP has both strengths and weakness that need to be discussed. As has been mentioned, VSP is a somewhat low-intensity, 1.5 hour meetings weekly over eight weeks. Retention was moderate compared to other studies as described above. Retention rates were not equal for all populations within the VSP program. Moving forward, we plan to assess the reasons that certain groups were less likely to partake in or complete the intervention through surveys and direct outreach. For example, male participants were more likely to drop out of the program than female participants, which has been seen in many other public health interventions [33]. Lower completion among participants with high A1c or glucose at baseline is also concerning. One possible explanation for this is that the structural barriers that led to poorer health at baseline, including challenges with transportation or childcare, also prevented participants from fully engaging in the class. In the future, we may want to evaluate some of these factors to see which drive patients to leave the class and discover ways to improve retention.

The catchment for this study is a convenience patient group at a single community clinic mainly utilized by the Hispanic patients of Caribbean, Central and South American origin and the surrounding neighborhood. Thus, the results may not be generalizable beyond similar clinics. Additionally, the evaluation protocol could be improved to measure and better understand VSP participant dietary and physical activity changes. Other comparable studies have included measures of physical activity [34], diet [26, 27], measures of inflammatory markers [30] and assessment of depressive symptoms [27–29]. An enhanced evaluation protocol might consider the addition of these factors along with assessment of medication adherence and other clinical compliance (e.g. self-monitoring of blood glucose, etc.). Longer-term follow up visits, such as at 6 months or 1 year after the program, may provide more robust data and allow us to determine the durability of the program's effects.

Another limitation is the fact that sessions often occurred in the evenings, so glucose and lipids had to be measured after 4 hours of fasting instead of the standard 8 hours. These point-of-care tests are a limitation in themselves; while previous studies have shown that values do not differ significantly [35] between fingerstick point-of-care and serum laboratory testing of glucose, a single glucose or cholesterol reading may not be representative of a patient's overall clinical picture. This may alter values slightly, but we were fortunately able to measure A1c as well, which provides a more reliable, longitudinal measure in the case of blood glucose.

A consistent concern for non-profit community organizations and free clinics is cost effectiveness and long-term sustainability. Overall, the costs of this program are relatively low, thanks in large part to the fact that CEHC is staffed mostly by volunteers. Direct costs, estimated at $5,000-$7,500 per class for 15–20 participants ($250-$500 per person), which includes *Navegante* salaries, printing, incentives, and clinical measurement supplies, and administrative costs include project coordination and financial management time. CEHC continuously seeks outside investment in this program, including collaboration with other health management initiatives through state and private entities that are geared towards low-income communities. By demonstrating the effectiveness and culturally appropriate nature of our program to funders and stakeholders, we are able to offer the program to participants who would not be able to afford a commercially available program.

The intervention curriculum focused on many aspects of obesity, lipids, blood pressure and diabetes, with guidelines for diet, physical activity, and medication compliance. An enhanced version might be strengthened with theory-based behavior change components and accompanying evaluation. Additionally, interventions in the literature have included home visits and individual meetings to set goals and provide one-on-one case management [27], which is a broader scope than the follow-up phone calls that take place in VSP. Other interventions have

also included coach-facilitated online group sessions or social media enhancements [36]; components that could be considered with or without *Navegante* presence for future enhancement of VSP. Future work should focus on increasing participant engagement throughout the intervention to strengthen the overall fidelity to the program design by improving the dose received. A randomized trial or other study design including a comparison or control group would also allow for better assessment of the intervention compared with outcomes of patients who might be motivated for change after a clinical diagnosis or discussion, but without this specific program for guidance.

The strengths of this study are also important to acknowledge. Participants anecdotally reported appreciation for the program, with particular enjoyment of the social connection and community-building that occurred, in addition to the individual attention of the *Navegantes*. The "social club" style seems to be a unique style of employing CHWs, which engaged participants with low-commitment emphasis, but also served in community-building. The style and client appreciation underscore the fact that this research is comprised of evaluation of a clinic-based program, not a funded research protocol. Other studies that also use CHWs describe some form of social engagement, but the non-classroom, fun activities may be unique to VSP. Also, VSP focused on a broad set of chronic disease risks, expanding beyond traditional focus on just diabetes or cardiovascular risk separately. While there are many external factors that may have impacted results, the enrollment of well over 600 participants for this small, clinic-based program is considerable, indicating the need and potentially strong benefit for VSP and the utility of further translational research.

It is important note that the participants in the Vida Sana program are among the lowest income individuals living in Providence, Rhode Island, and they face numerous structural inequities that negatively impact their health and access to health care. They do not speak English as their first language, work multiple jobs, lack reliable sources of food and transportation, and live in crowded multi-tenant housing. Their ability to improve their own health is limited by these circumstances, and community interventions like ours can unfortunately not address the many systemic factors at play. However, it appears that despite substantial barriers to improving health, providing participants with new tools and information may empower them to reclaim some control over their health. These acquired skills could have a long lasting effect.

In sum, the VSP shows encouraging improvements in clinical metabolic parameters for participating members of the CEHC community, especially given the limited intensity and scale of the intervention. Future research may continue this important work by carefully expanding the intervention and evaluation, but without losing the strong community engagement. Future research should include assessment of cost effectiveness, strengthening retention rates in certain sub-populations (including men and those with high baseline A1c), further assessment of diet, physical activity and medication compliance, and exploration of the social connection between participants and *Navegantes*.

## Acknowledgments

The authors wish to acknowledge all of the *Navegantes* who participated in conducting the Vida Sana program.

## Author Contributions

**Conceptualization:** Patricia Markham Risica, Anne S. De Groot.

**Data curation:** Katherine L. Barry.

**Formal analysis:** Patricia Markham Risica, Katherine L. Barry.

**Investigation:** Anne S. De Groot.

**Resources:** Anne S. De Groot.

**Writing – original draft:** Meghan L. McCarthy.

**Writing – review & editing:** Patricia Markham Risica, Susan P. Oliverio, Kim M. Gans, Anne S. De Groot.

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
