## [Decision Letter · Decision Letter 0]

15 Dec 2020

PONE-D-20-15540

Clinical Outcomes of a community clinic-based lifestyle change program for prevention of metabolic syndrome: Results of the ‘Vida Sana/Healthy Life’ program

PLOS ONE

Dear Dr. Risica,

Thank you for submitting your manuscript to PLOS ONE. After careful consideration, we feel that it has merit but does not fully meet PLOS ONE’s publication criteria as it currently stands. Therefore, we invite you to submit a revised version of the manuscript that addresses the points raised during the review process.

The authors present an well-written, interesting, and timely study describing the outcomes observed from their implementation of a community-centered group care strategy led by Hispanic CHW’s.  The analysis is thorough and merits publication to provide guidance to other programs trying to implement similar programs.   This should be possible after addressing the comments made by the reviewers.   Please pay special attention to the comments about the analysis. 

We look forward to receiving your revised manuscript.

Kind regards,

Sonak D. Pastakia

Academic Editor

PLOS ONE

Journal Requirements:

2. In your Methods section, please provide additional information regarding the program. If materials, methods, and protocols are well established, authors may cite articles where those protocols are described in detail, but the submission should include sufficient information to be understood independent of these references (https://journals.plos.org/plosone/s/submission-guidelines#loc-materials-and-methods); for example, the role of "navegantes" needs to be explained in more detail.

Moreover, please ensure you have provided sufficient details to replicate the analyses such as:

a) the recruitment date range (month and year),

b) a description of any inclusion/exclusion criteria that were applied to participant recruitment,

c) a table of relevant demographic details,

d) a statement as to whether your sample can be considered representative of a larger population,

e) a description of how participants were recruited, and

f) descriptions of where participants were recruited and where the research took place.

- https://www.sciencedirect.com/science/article/pii/S2451865418301005?via%3Dihub

 The text that needs to be addressed involves some sentences of the Introduction.

In your revision ensure you cite all your sources (including your own works), and quote or rephrase any duplicated text outside the methods section. Further consideration is dependent on these concerns being addressed.

Additional Editor Comments:

Introduction

Minor comment – pg 5, line 88, If this is data is from years 2-5 data, why is the duration of follow-up so short? Would have loved to see the persistence of these changes since it seems like it is just a 12 week intervention and then it is over. Is that the case? Is there not demand from the group members to continue these meetings after

Methods

Minor comment – Can you briefly mention how the Navegantes and the study activities integrate with the rest of the clinical team?

Minor comment – pg 7 , line 138, are there any guidelines which suggest that a fasting blood sugar with four hours of fasting and using a point of care glucose meter is adequate for monitoring individual patient care? If not, please highlight this as a limitation as I can understand your reasons for using a point of care test. It is worth noting that there are accuracy concerns with point of care meters.

Minor comment – pg 9, line 171, Slightly confused about the imputation of missing values. Did you do a calculation to predict missing data or did you just carry the last value forward. I think your wording is clear but just wanted to confirm. My one concern with this is that with >30% lack of completion, this might really impact the analyses

Minor comment – Do you have any comment on the sample size considerations for this intervention? What clinically significant differences in outcomes were you designing your study to assess? It seems like there were many significant differences noted but some of minimal clinical significance.

Stylistic comment - I would have like to also see some kind of composite outcome if you could imagine one for this work. I can’t think of one outside of maybe the AHA CV risk score but would be a nice thing to add for future studies.

Results

Minor comment - Pg 12 second paragraph. Can you comment further in the discussion about why people with higher a1c or glucose were less likely to complete the intervention? Do you have any sense of the number of other appointments they had which may explain why they skipped this appointment?

Minor comment – page 13, table 2 – what is in the parentheses for knowledge assessment?

Discussion

Stylistic comment - can you talk about the costs of this program?

Minor Comment - Past studies have often shown that men tend to have less uptake in these kinds of groups. Can you comment on that in the discussion or any adaptations you would suggest going forward.

Major comment - Pg 18, 4th paragraph – There is a significant typo – “reduction of 8.3 mg/dL in A1c was observed”. I’m assuming this should be fasting blood sugar instead of A1c.

Reviewers' comments:

Reviewer's Responses to Questions

**Comments to the Author**

1. Is the manuscript technically sound, and do the data support the conclusions?

Reviewer #1: Partly

2. Has the statistical analysis been performed appropriately and rigorously? 

Reviewer #1: Yes

3. Have the authors made all data underlying the findings in their manuscript fully available?

Reviewer #1: No

4. Is the manuscript presented in an intelligible fashion and written in standard English?

Reviewer #1: Yes

5. Review Comments to the Author

Reviewer #1: General Comments

Overall this is a very well written paper on an interesting topic – a community clinic-based lifestyle change program for an underserved, largerly Spanish-speaking community in Rhode Island. The authors should be commended for their very clear description of the methods and the results sections in particular. This manuscript would certainly add to the literature. However, I do have some concerns that the authors’ conclusions do not reflect their findings. In particular, although the changes in glucose, cholesterol and blood pressure were statistically significant for all participants, they do not seem clinically impressive.

Additionally, the baseline knowledge of participants seemed quite high, and it is likely that systemic problems such as access to inexpensive healthy foods, increased stress levels, and limited time available greatly impact disease management and diet/exercise modification. It would be great to see the authors incorporate some discussion about the fact that these types of community interventions unfortunately are unable to tackle the larger structural inequities at play and that relatively high baseline knowledge levels among participants potentially indicates that literacy is likely only one small factor that needs to be addressed.

Other Comments

1) Title: The title implies that this program is for ‘prevention’ of metabolic syndrome. However, it seems that some participants already had metabolic syndrome at baseline. Please clarify.

2) Lines 74-77: Recommend adding a brief explanation of why these behavioral risk factors are higher in Hispanics compared to non-Hispanic whites – i.e. systems of oppression/structural inequities upstream likely leading to racial/ethnic behavioral differences. Might be interesting to add additional evidence from the Healthy Americas Survey such as question HA-13 regarding cost of fresh fruits and vegetables, how do whites vs Hispanics respond to this question?

3) Line 88: please include the years during which the study took place since readers will not know when years 2-5 were.

4) Lines 103-104: Since recruitment extends to community members interested in the program, I do wonder if this incorporates a selection bias? Did the authors compare outcomes between those who opted into the program to others?

5) Lines 107-111: Would be helpful if the authors could comment in the discussion on how they envision this program being sustainable given that it seems to cost >$30/participants not including any cost of the materials, on-site child care services, and Navegantes. I am curious if any of the participants mentioned using the money to buy healthy foods? If so, this should be mentioned as a potential driver of behavior change as well.

6) Line 118: please correct reference 10 as it is written ‘in submission’ and update the reference appropriately.

7) Lines 138-143: For clinical purposes, fasting glucose and cholesterol are usually measured after 8 hours of fasting. Therefore, the results in this study may not correspond to true clinical fasting glucose or cholesterol measurements and should be mentioned as a limitation or clarified if a different guideline was used.

8) Line 164: It would be helpful to understand why SBP of 130 and DBP of 80 were selected as the high-risk blood pressure cut off and provide a reference for the guideline used.

9) Line 169: Given that a parametric test was utilized, please comment regarding the distribution of the population. If the population was not normally distributed, were corresponding non-parametric statistical analyses performed for comparison to ensure consistency in statistical significance?

10) Table 1: This is an excellent table. It is interesting that it seems participants who did not complete the program were more likely to have uncontrolled diabetes with A1Cs>7.0. I would be interested to know if these participants were more likely to be uninsured or if they had other socioeconomic constraints that both impacted their ability to manage their DM as well as attend classes. The authors could consider expanding on this in the discussion if this data is available.

11) Table 3: This is also a great table. However, although most of these variables are statistically significant, they do not seem clinically impressive, especially in the all participants group. I think that this point needs to come across in the discussion and needs to be strongly considered in terms of whether or not this program is truly impacting all participants’ metabolic syndrome. For high risk participants, the A1C decrease of 0.39 in a two month period as well as the systolic BP drop by 10mmHg should be highlighted in the discussion and perhaps the authors might consider that this program should target participants who are highest risk in the future.

12) Paragraph 7 discussion, “Changes in indicators of diabetes….”: In the sentence, Among those beginning VSP with elevated blood glucose levels, a reduction of….in ‘A1C’ I believe should be ‘glucose’.

13) Overall in the discussion, it is unclear how much information was known about participants’ chronic disease management and if many of them were already on or potentially started on medications during the study. It would be helpful if the authors could include more information about this in the methods since participants who opt to join this type of program may also be opting to attend their clinical visits and participate in medication management.

14) Regarding lipid levels, it would be helpful to understand why the authors chose cholesterol levels rather than calculating ASCVD risk for each participant as it seems like the information needed to calculate this score was available.

15) The paragraph about strengths is excellent. If the authors have any additional qualitative data that were collected about the social connection and individual connection with Navegantes, it would be worthwhile to add this formally to the results. If this program is ongoing, it would be very interesting to conduct a qualitative study with participants and Navegantes to further explore this social aspect and the impact this played in suspected behavioral change.

16) The sentence, “The reach of well over 600 participants for this small….indicating the need and potentially very strong benefit for VSP” is strongly worded. This study does demonstrate a significant increase in several vital signs and lab variables, however, as the authors mention, the change may very well have been associated with medications or other factors that were not accounted for. Would recommend altering this to reflect results.

6. PLOS authors have the option to publish the peer review history of their article (what does this mean?). If published, this will include your full peer review and any attached files.

Reviewer #1: No

---

## [Author Response · Author response to Decision Letter 0]

12 Feb 2021

A full table of responses to every reviewer comment has been submitted with this revision. We appreciate the time and attentioni paid by the PLOS reviewers.

---

## [Decision Letter · Decision Letter 1]

1 Mar 2021

Clinical Outcomes of a community clinic-based lifestyle change program for prevention and management of metabolic syndrome: Results of the ‘Vida Sana/Healthy Life’ program

PONE-D-20-15540R1

Dear Dr. Risica,

We’re pleased to inform you that your manuscript has been judged scientifically suitable for publication and will be formally accepted for publication once it meets all outstanding technical requirements.

Kind regards,

Sonak D. Pastakia

Academic Editor

PLOS ONE

Additional Editor Comments (optional):

Reviewers' comments:

Reviewer's Responses to Questions

**Comments to the Author**

1. If the authors have adequately addressed your comments raised in a previous round of review and you feel that this manuscript is now acceptable for publication, you may indicate that here to bypass the “Comments to the Author” section, enter your conflict of interest statement in the “Confidential to Editor” section, and submit your "Accept" recommendation.

Reviewer #1: All comments have been addressed

2. Is the manuscript technically sound, and do the data support the conclusions?

Reviewer #1: Yes

3. Has the statistical analysis been performed appropriately and rigorously? 

Reviewer #1: Yes

4. Have the authors made all data underlying the findings in their manuscript fully available?

Reviewer #1: Yes

5. Is the manuscript presented in an intelligible fashion and written in standard English?

Reviewer #1: Yes

6. Review Comments to the Author

Reviewer #1: Excellent revision. Thank you for addressing all of the comments thoughtfully.

Please fix line 378 in discussion - seems that this sentence is incomplete.

7. PLOS authors have the option to publish the peer review history of their article (what does this mean?). If published, this will include your full peer review and any attached files.

Reviewer #1: No

---

## [Editor Report · Acceptance letter]

16 Mar 2021

PONE-D-20-15540R1 

Clinical Outcomes of a community clinic-based lifestyle change program for prevention and management of metabolic syndrome: Results of the ‘Vida Sana/Healthy Life’ program 

Dear Dr. Risica:

I'm pleased to inform you that your manuscript has been deemed suitable for publication in PLOS ONE. Congratulations! Your manuscript is now with our production department. 

Kind regards, 

on behalf of

Dr. Sonak D. Pastakia 

Academic Editor

PLOS ONE